# Studies on the hysteresis of trunk muscles— Muscular specificities must be taken into account

**Christoph Anders** *, **Leon Rosenow**

Division of Motor Research, Pathophysiology and Biomechanics, Experimental Trauma Surgery, Department for Hand, Reconstructive, and Trauma Surgery, Jena University Hospital, Friedrich-Schiller-University Jena, Jena, Germany

¤ Current address: Stadtspital Waid, Zürich, Switzerland

* Christoph.anders@med.uni-jena.de

**Data Availability Statement:** All relevant data are within the manuscript and its Supporting Information files.

**Funding:** The author(s) received no specific funding for this work.

## Abstract

Hysteresis refers to a physical phenomenon in which the response or state of a system depends on both the input variable and its history. Hysteresis phenomena are also observed in biological systems and have been described for the sensorimotor system. The aim of the present study was to determine whether hysteresis phenomena can also be detected in trunk muscles during isometric load-varying situations. To this end, 40 healthy individuals (19 women) were subjected to isometric tests, where the applied load was systematically altered by complete rotations of the entire body in the Earth's gravitational field. The study was conducted with 25%, 50%, and 75% of the upper body weight. Additionally, we varied the starting point (forward tilt and backward tilt) and the direction of rotation. The activity of a total of six trunk muscles was recorded using surface EMG (sEMG). The sEMG amplitudes were compared between the phases of increasing and decreasing load levels for each test situation. Hysteresis behavior was observed in all examined muscles, with the movement half-phase with increasing load showing higher amplitudes than the half-phase with decreasing load. However, this was consistently verifiable only for the multifidus muscle. For the abdominal muscles, the longissimus, and the iliocostalis muscle, the occurrence of hysteresis depended on the starting position: it could only be demonstrated if the starting point was chosen to correspond with the muscles' main force direction. Thus, only the multifidus muscle exhibits a situation-independent hysteresis, whereas all other examined trunk muscles only show this phenomenon if subjected to load already at a loading situation. This indicates a physiologically determined functional weakness for load impacts on primarily unloaded muscles, posing a potential injury risk.

## Introduction

The concept of hysteresis is virtually omnipresent, as this phenomenon can be observed in various fields, including physical and technical domains, economics, and also in physiological

**Competing interests:** he authors have declared that no competing interests exist.

processes. The term describes a system behavior in which a response lags behind its cause. This means that the response or state of the reacting system depends not only on the state of the input variable but also on its history. In physiology, a well-known example of this is the pressure-volume curve of the lungs. Here, deviations in flow rates for the same pressures between inspiration and expiration are characteristically altered in pathologies such as emphysema [1].

Similar evidence can be found in the musculoskeletal system. For example, the viscoelastic creep phenomenon [2, 3], where the sarcomere length increases following prior stretching. According to the well-known force-length relationship governing muscle contraction, this leads to a reduction in force production capacity [4]. As a result, the risk of injury increases because less counterforce can be generated if external forces act on the system. This process is reversible once the triggering cause, i.e. the stretching is terminated. Evidence for this has been demonstrated both in cadaver studies [5, 6] and in physiological experiments [3, 7, 8].

A similar phenomenon can be observed for the different types of muscle contraction. During concentric contraction, more muscular effort and, therefore, higher neural activation is required to overcome the same load compared to an eccentric contraction of equal load [9]. The reason for this lies in the physiology of the basic contraction cycle, where shortening occurs actively with energy expenditure, while the actin-myosin interaction remains fixed without additional energy consumption [1]. However, experimental conditions seem to play a crucial role here, as these differences were not observed during isometric contractions of the calf muscles [10].

With respect to the trunk muscles, whose adequate function ensures the integrity of the spine, beyond the aforementioned basic physiological phenomena an additional factor has to be considered: adequate motor control, which can be altered by subfailure injuries and must therefore additionally be taken into account [11, 12].

Given this context, the present study aimed to investigate how the activation of trunk muscles is altered under isometric conditions with continuously changing load demands. Therefore, the investigation was designed as a physiological investigation to identify potential hysteresis effects, which, if demonstrable, could also have practical implications. Such effects would be particularly relevant in scenarios involving sudden forces acting on the trunk, possibly altered by the muscles functional baseline situation. Specifically, we sought to determine whether hysteresis phenomena could be observed. To this end, a group of healthy individuals was subjected to defined and continuously changing load applications of their trunk muscles. Based on existing preliminary findings, which suggest significantly different response patterns between the abdominal and back muscles, we expected to observe different results between these regions—anticipating pronounced hysteretic behavior in the abdominal muscles and little to no hysteresis in the back muscles.

## Methods

Forty healthy individuals (19 women) volunteered to participate in this study. Participants were recruited through electronic announcements and personal invitations among university staff and students. The study was conducted in October 2011. All participants provided written informed consent for voluntary participation. The study was approved by the Ethics Committee of the Friedrich-Schiller-University (3021-01/11, date: 03/07/2011). Prior to the investigation, a general orthopedic examination was conducted to identify any notable findings.

The following exclusion criteria were defined: age under 18 or over 60 years, acute or chronic back pain, previous spinal or trunk surgeries, spinal deformities, osteoarthritis of major joints, any use of pain medication, known pregnancy or heart conditions, and a BMI $>30$ kg/m$^2$.

## Investigation procedure

Participants were positioned standing in a computer-controlled testing and training device (CTT Centaur®, BfmC, Leipzig, Germany), with their lower body fixed and their upper body remaining free to move (see Fig 1). The device comes with a shoulder harness that is equipped with force sensors in x and y axes. Data from these sensors are displayed on a feedback monitor directly in the participants' line of sight. The monitor features a movable point within a crosshair. If a participant applies any force to the shoulder harness, the point moves away from the center of the crosshair in the respective direction. Hence, when the point is located in the center of the crosshair, the participant stands in upright posture. Regardless of their spatial position, participants were required to maintain upright posture, which was controlled via the feedback monitor. Any deviations from upright posture could thus be identified and corrected if necessary. The device has already been used in several other studies [13, 14], demonstrating its reliable application [15, 16].

The device performs whole-body tilts, so the participant in the device does not need to make any movements and thus does not need to leave the upright posture. Tilt angles can be

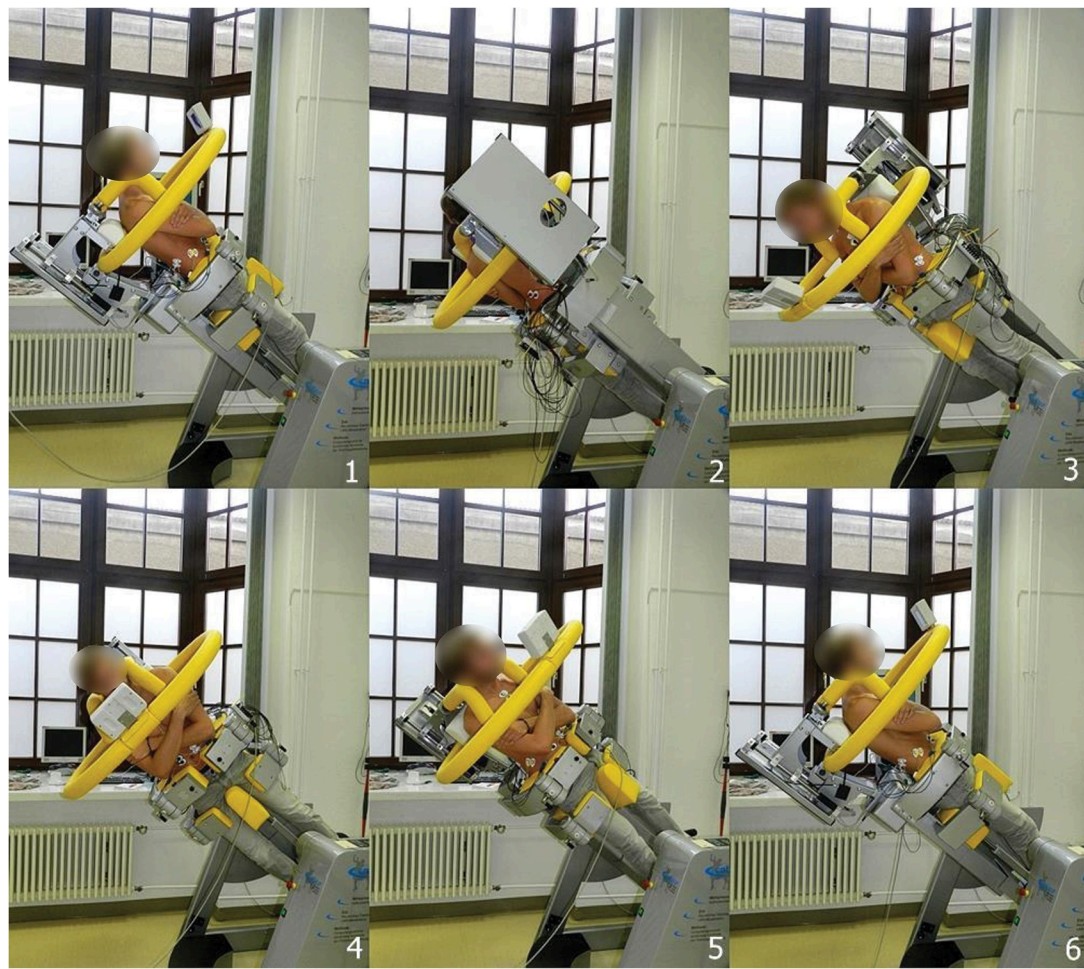

**Fig 1. Presentation of the test setup exemplarily at 49˚ tilt angle, starting position 180˚ (backward tilt), rotation direction counterclockwise (Subfigures 1–6).** Note that the lower body is fixed while the upper body remains free to move. Also, notice the feedback monitor placed directly in the participant's line of sight to ensure adherence to upright posture during exercise execution.

freely selected, ranging from 0˚ (no tilt) to 90˚ (horizontal position). Additionally, the device can set defined rotational angles or perform continuous rotations around the body axis within a defined range. For this study, a combination of both options was used. Once positioned in the device, participants crossed their arms in front of their chest. The experiments were conducted with tilt angles of 15˚, 30˚, and 49˚, corresponding to 25%, 50%, and 75% of the upper body weight (UBW). Each exercise started either at 0˚ (forward tilt) or at 180˚ (backward tilt) rotation angle. To avoid acceleration or deceleration effects, each exercise started and ended with a lead-in and lead-out of 20˚. From these starting positions, a 400˚ rotation was performed, from which the complete 360˚ rotation was analyzed either clockwise or counterclockwise. The rotation speed was 18˚/s, so a complete 360˚ rotation took 20 seconds. Therefore, for each tilt angle, a total of four complete rotations were performed: clockwise and counterclockwise rotations, with starting points at 0˚ and 180˚. Thus, a total of 12 test scenarios were examined per participant.

## sEMG measurement

During the investigation, the activity of six superficial trunk muscles was recorded using bipolar surface EMG (sEMG): three abdominal muscles and three back muscles (see Table 1), as well as an additional ECG channel for distinct identification of heart activity. The electrode placement followed international recommendations [17–19]. The skin at the respective positions was first prepared with abrasive paste (Epicont, GE Healthcare, Germany), and hair was shaved if necessary. Electrodes were then attached (H93SG, Arbo, Neustadt, Germany; electrode diameter 1.6 cm, inter-electrode distance 2.5 cm). The measured sEMG signal was amplified (gain: 1000, input impedance: 1200 GΩ, noise level: < 1 μV, 10–700 Hz, RC filter 1st order, Biovision, Wehrheim, Germany), analog-to-digital converted (2048/s, Tower of Measurement (ToM), 24-bit resolution at ± 5V: 0.6 nV/Bit, anti-aliasing filter at 1024 Hz, DeMeTec, Langgöns, Germany), collected (ATISArec, GJB Datentechnik Ilmenau, Germany) and stored for offline processing. All measured signals were checked for baseline noise, mains hum, and signal validity during the measurement. Electrodes or amplifiers were replaced if necessary.

## Data processing and statistics

Data were band-pass filtered (20–400 Hz) to eliminate movement artifacts and low-frequency components of heart activity [20]. QRS artifacts were removed using a template algorithm

**Table 1. Investigated muscles and electrode positions.**

| Muscle | Electrode position und alignment |
|---|---|
| M. rectus abdominis | Caudal electrode at navel height, 4cm lateral midline, vertical |
| M. obliquus internus | Medial of inguinal ligament, at ASIS, height, horizontal |
| M. obliquus externus | Cranial electrode just below lower margin of costal arch, on line from there towards contralateral pubic tubercle |
| M. iliocostalis | cranial electrode at L2 height, medial from line from PSIS to lowest point of the costal arch |
| M. multifidus | caudal electrode at L5 height, 1 cm medial and parallel to line between PSIS and L1 |
| M. longissimus | caudal electrode at L1 height, over palpable bulge of muscle (approx. 2 fingers lateral from midline), vertical |
| EKG | Along heart axis, above heart |

ASIS: anterior superior iliac spine, PSIS: posterior superior iliac spine, L: palpable spinous process of lumbar vertebra and position

[21]. Additionally, possible interference from the power supply was eliminated with a 50 Hz notch filter. The processed data were quantified as root mean square (RMS) within a moving time window of 50 ms. Data were further normalized to a complete 360˚ rotation with a resolution of 0.5%, meaning each data point represents the average RMS amplitude for 1.8˚ rotation. These data were then used for the final analysis. To improve comparability, the two half-phases of a complete rotation are always presented at the same side of the respective diagrams. Data are displayed as Cartesian coordinates, with scaling based on amplitude and angular data, thus represented as arbitrary units (a.u.).

When calculating the sample size, an effect size of 0.5 was assumed for a two-sided hypothesis. Assuming a power of 0.8 for the testing of paired samples, a sample size of 34 was considered sufficient. Statistical comparisons to identify hysteresis effects were performed for corresponding angular positions between the respective movement half-phases. As the data for the 40 participants assumed to follow a normal distribution, we initially applied the parametric t-test for dependent samples. These initial statistical results were then corrected for multiple comparisons using the Bonferroni-Holm method [22–24]. Remaining significant values ($p < 0.05$ after correction) are displayed in the diagrams according to their angular positions.

## Results

Systematic rotation direction-dependent findings regarding the achieved amplitude values or the occurrence of hysteretic differences could not be identified, so all subsequently reported results apply to both rotation directions.

### M. rectus abdominis (RA)

For the RA, indications of hysteretic behavior were found, although a consistent statistical confirmation was not achieved (Fig 2). The muscle activation behavior varied depending on the starting point. At the starting point of 0˚ (forward tilt), no differences between the movement half-phases were observed at 25% and 50% UBW, but at 75% UBW, higher amplitudes were noted for the half-phase with increasing load, which were detectable occasionally. At the starting point of 180˚ (backward tilt), higher amplitudes for the half-phase with increasing load were consistently observed regardless of the tilt angle, though systematic differences could not be demonstrated. Nevertheless, the observed amplitude values at 0˚ and 180˚ rotation angles were virtually identical. Peak amplitudes were achieved for 180˚ rotation angle (backward tilt), with group means of approximately 12 a.u. at 25% UBW, 40–50 a.u. at 50% UBW, and 100–130 a.u. at 75% UBW.

### M. obliquus internus (OI)

The OI exhibited a very similar behavior to that of the RA: the half-phases with increasing load showed higher amplitudes compared to the half-phases with decreasing load (Fig 3). However, for the starting point of 0˚ (forward tilt), this could only be observed and confirmed at 75% UBW. Corresponding effects were visible for the starting point of 180˚ (backward tilt) regardless of the tilt angle, but again systematic differences could not be confirmed. Peak amplitudes were achieved for 180˚ rotation angle (backward tilt), with group means of approximately 25 a.u. at 25% UBW, 70–80 a.u. at 50% UBW, and 140–160 a.u. at 75% UBW.

### M. obliquus externus (OE)

Again, for the starting point of 0˚ (forward tilt), visible and systematic amplitude differences between the two movement half-phases were only observed at 75% UBW, with the half-phase

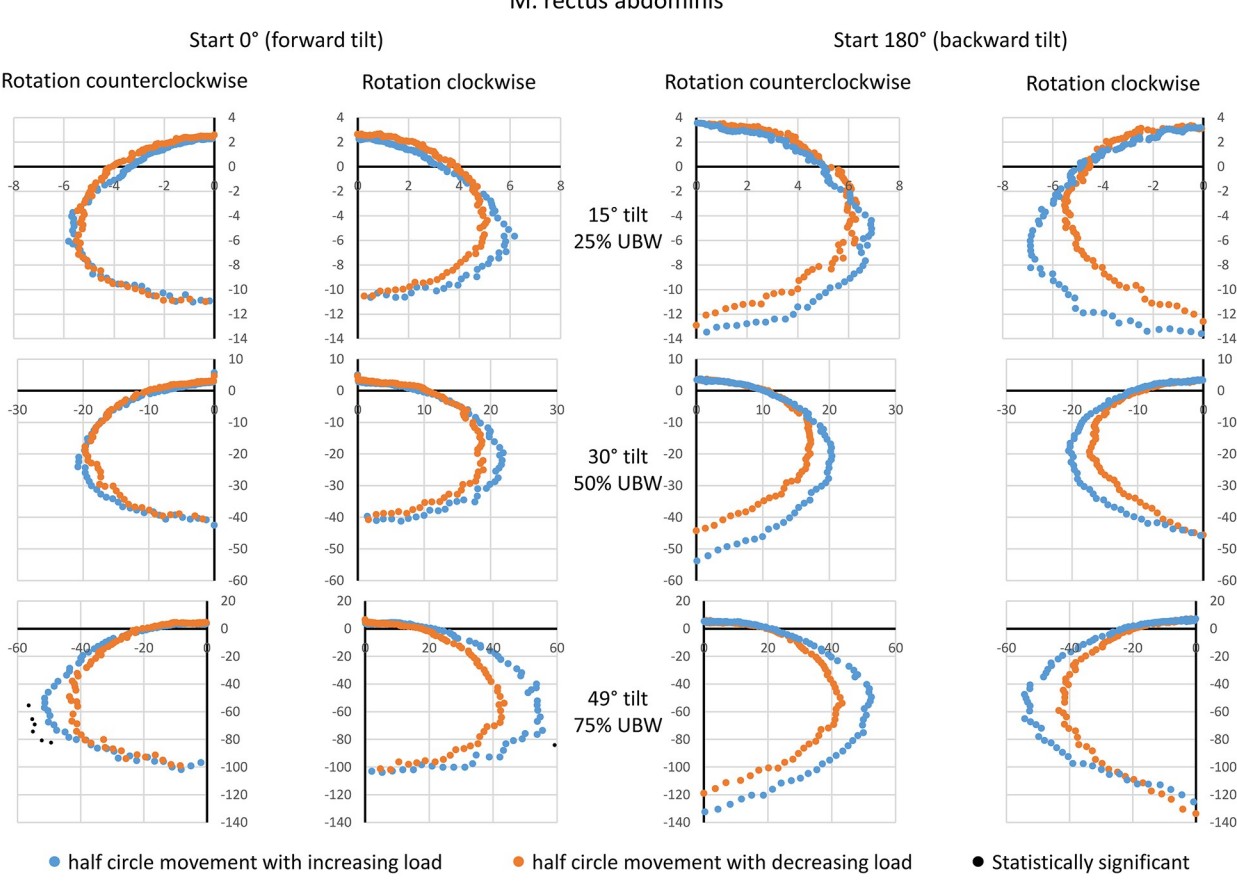

**Fig 2. Mean amplitude values during a complete 360˚ rotation for the rectus abdominis muscle.** Data are presented based on the averaged amplitude values from both sides of the body, expressed in arbitrary units. The values for the movement half-phase with decreasing load are mirrored to enable a direct comparison between both movement half-phases. Statistical significant differences (p < 0.05, after Bonferroni-Holm correction [22]) are also displayed. UBW: upper body weight.

involving increasing load showing higher amplitude values (Fig 4). For the starting point of 180˚ (backward tilt), the half-phase with increasing load exhibited visibly and systematically higher amplitude values compared to the half-phase with decreasing load, starting already at 25% UBW. Peak amplitudes were achieved for 180˚ rotation angle (backward tilt), with group means of approximately 20 a.u. at 25% UBW, 40–45 a.u. at 50% UBW, and 80 a.u. at 75% UBW.

## M iliocostalils (ICO)

For the ICO, visible and systematic amplitude differences between the movement half-phases were only detected for the starting point of 0˚ (forward tilt, Fig 5). Again, the half-phase with increasing load exhibited higher amplitude values compared to the half-phase with decreasing load. For the starting point of 180˚ (backward tilt), no visible or detectable hysteresis effects were observed. Additionally, it was noted that similarly high amplitudes were observed regardless of the starting point, at 0˚, 180˚, and also at ± 90˚ rotation angles. The observed peak amplitude values averaged approximately 8 a.u. at 25% UBW, 15 a.u. at 50% UBW, and 20–30 a.u. at 75% UBW.

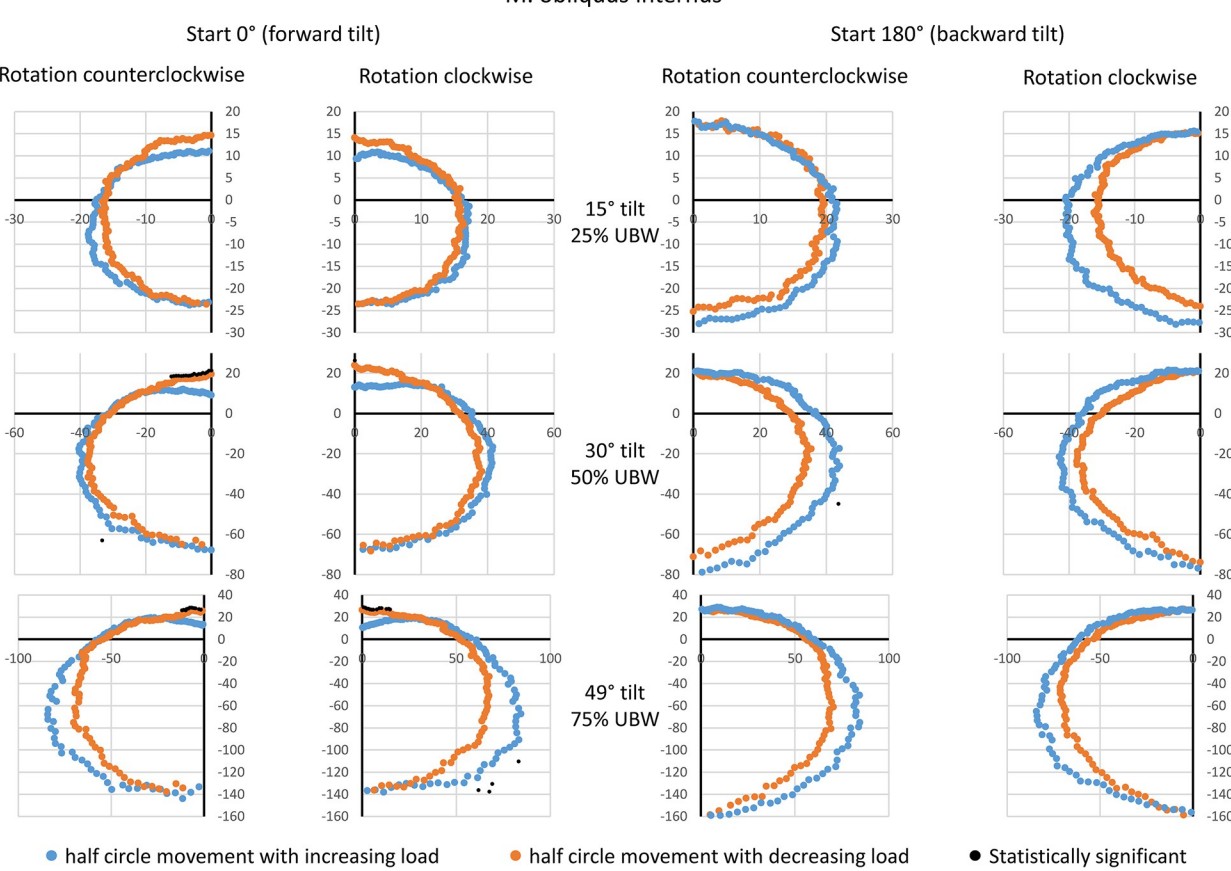

**Fig 3. Mean amplitude values during a complete 360˚ rotation for the internal oblique muscle.** Data are presented based on the averaged amplitude values from both sides of the body, expressed in arbitrary units. The values for the movement half-phase with decreasing load are mirrored to enable a direct comparison between both movement half-phases. Statistical significant differences (p < 0.05, after Bonferroni-Holm correction [22]) are also displayed. UBW: upper body weight.

## M. multifidus (MF)

Visible and systematic hysteresis effects were consistently detectable for MF, regardless of the starting point, tilt angle, and rotation direction. The half-phase with increasing load always exhibited higher amplitude values compared to the half-phase with decreasing load (Fig 6). Peak amplitudes were achieved for 0˚ rotation angle (forward tilt), with group means of approximately 35 a.u. at 25% UBW, 50–55 a.u. at 50% UBW, and 60–70 a.u. at 75% UBW.

## M. longissimus (LO)

For LO, any visible and detectable hysteresis was only observed for the starting point of 0˚ (forward tilt, Fig 7). Again, the half-phase with increasing load showed higher amplitude values compared to the half-phase with decreasing load. For the starting point of 180˚ (backward tilt), no general hysteresis was detectable. Systematic amplitude differences could only be found around the starting or ending point at 0˚ rotation angle: here the half-phase with decreasing load exhibited higher amplitude values. Peak amplitudes were achieved for 0˚ rotation angle (forward tilt), with group means of approximately 20–25 a.u. at 25% UBW, 30–35 a.u. at 50% UBW, and 40–50 a.u. at 75% UBW.

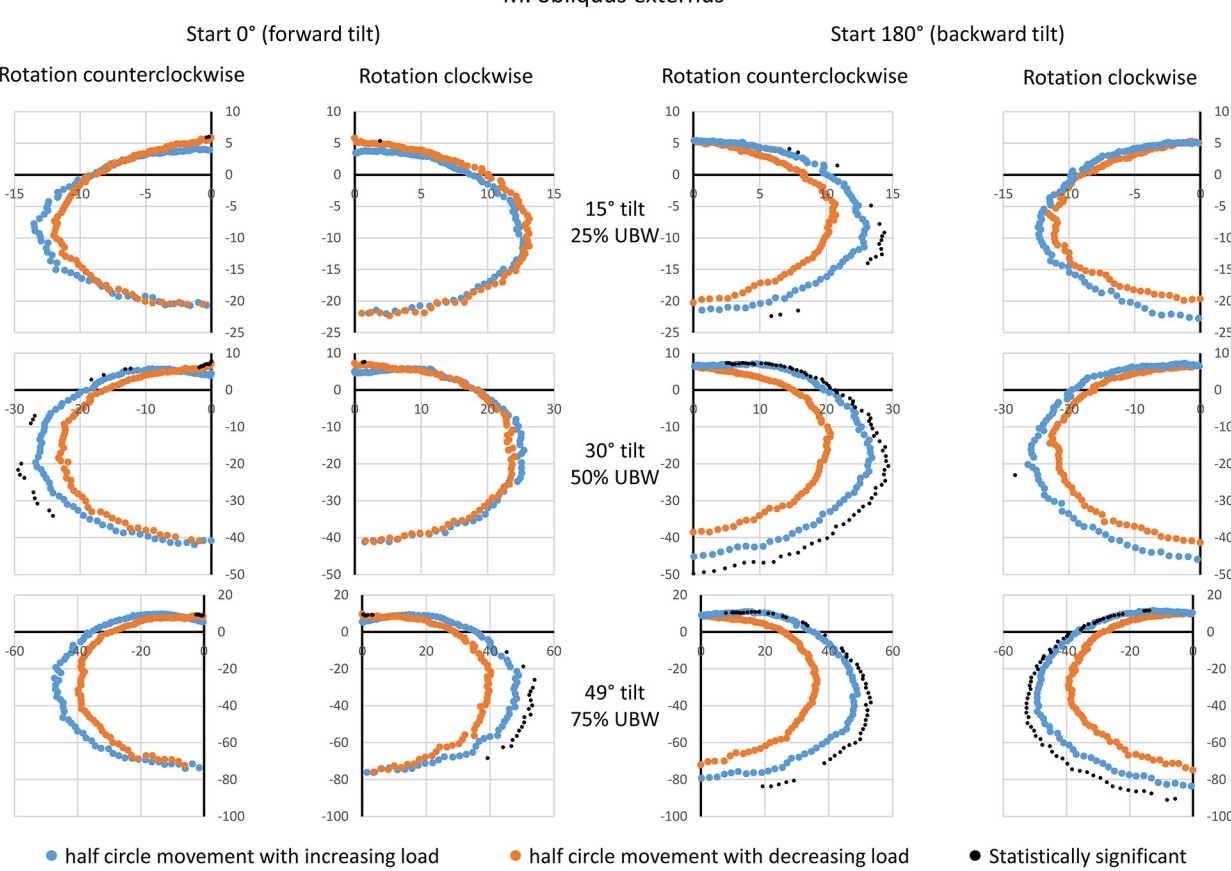

**Fig 4. Mean amplitude values during a complete 360˚ rotation for the external oblique muscle.** Data are presented based on the averaged amplitude values from both sides of the body, expressed in arbitrary units. The values for the movement half-phase with decreasing load are mirrored to enable a direct comparison between both movement half-phases. Statistical significant differences (p < 0.05, after Bonferroni-Holm correction [22]) are also displayed. UBW: upper body weight.

## Discussion

In this study, a specific loading scenario was applied to detect possible hysteresis behavior in trunk muscles: participants were subjected to isometric tests with continuously varying load levels by performing complete rotations at defined tilt angles within the gravitational field. The tilt angles were set to apply loads corresponding to 25%, 50%, and 75% of UBW on the torso. Additionally, both the direction of rotation and the starting point of the measurement were varied. An important aspect for data interpretation is that the averaged amplitude values from both sides of the body were analyzed—this reflects the combined load effect of both muscles, essentially serving as an electrophysiological activity vector. Thus, any expected lateral differences in response to the testing conditions are not the focus of this investigation.

For all the examined muscles, hysteresis behavior could be identified, with higher amplitude values for the half-phase with increasing load compared to the half-phase with decreasing load. However, there were significant differences between the muscles. For all abdominal muscles, hysteresis at the starting point of 0˚—which is opposite to the main force direction [25]—was only detectable for the highest applied tilt angle of 49˚, corresponding to 75% UBW. For lower load levels, no systematic influence of load-level change could be demonstrated at this starting point. The situation was quite different when the starting point was set at 180˚, ideally aligned

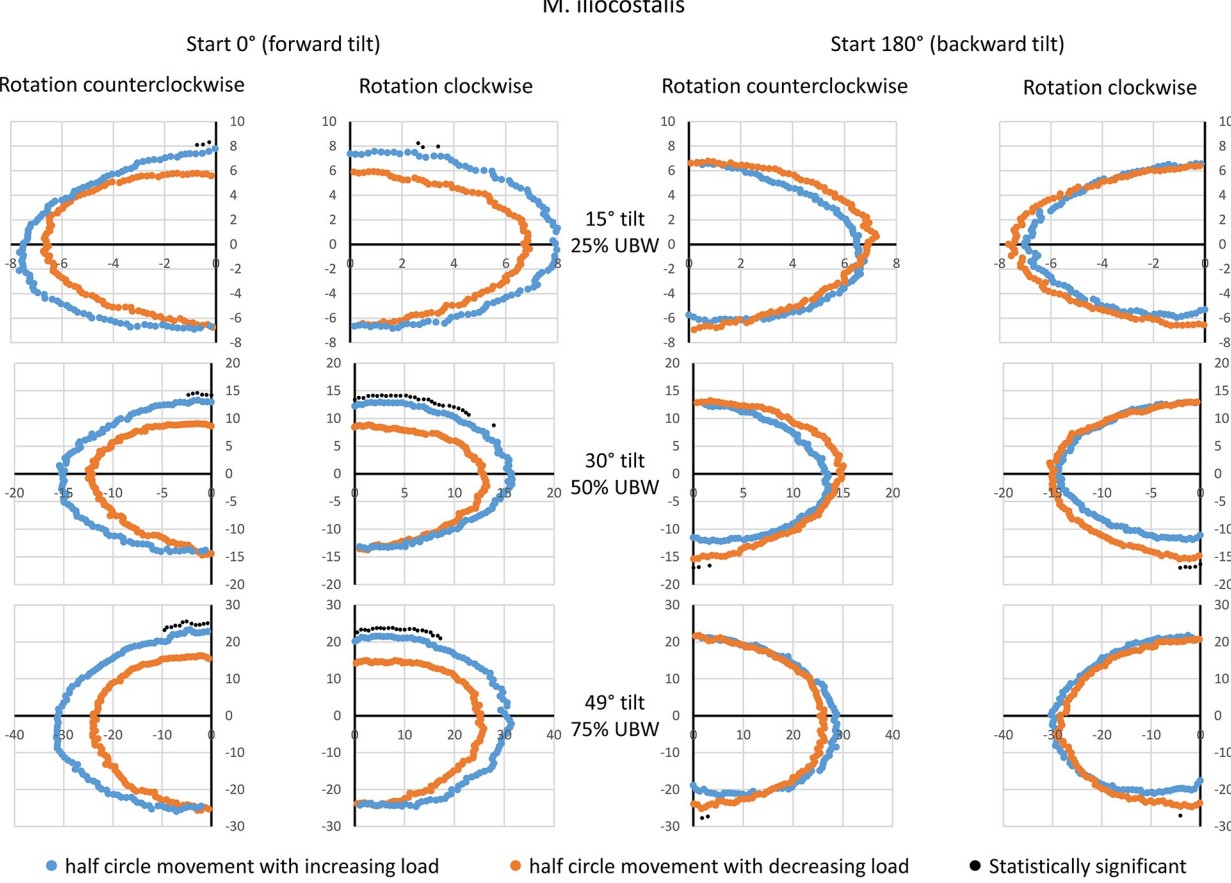

**Fig 5. Mean amplitude values during a complete 360˚ rotation for the iliocostalis muscle.** Data are presented based on the averaged amplitude values from both sides of the body, expressed in arbitrary units. The values for the movement half-phase with decreasing load are mirrored to enable a direct comparison between both movement half-phases. Statistical significant differences (p < 0.05, after Bonferroni-Holm correction [22]) are also displayed. UBW: upper body weight.

with the main force direction [25]. Here, hysteresis behavior was consistently visible regardless of the load levels. However, statistical testing largely failed to demonstrate systematic differences. Only for OE the visible difference between the two movement half-phases was systematically detectable.

Regarding statistical verification, it is important to note that the initially applied tests were subjected to the Bonferroni-Holm correction to avoid the accumulation of Type I errors (alpha errors) [22–24]. Although this method can be considered considerably less conservative than the classical Bonferroni correction, initially the lowest p-values had to be multiplied by 100. This means that an uncorrected p-value of < 0.0005 would have been required for a single value to achieve the set global significance level of p < 0.05. Such low p-values are challenging to obtain, even with substantial deviations. Therefore, we also calculated effect sizes (ES), which confirmed the visible hysteresis as relevant far more frequently than the statistical significance test (ES > 0.5, see S1–S6 Figs). This approach provided a more nuanced understanding of the data, highlighting the relevance of observed hysteresis effects that might not meet the strict criteria required by the Bonferroni-Holm correction.

In contrast, the investigated back muscles revealed a heterogeneous pattern, with each muscle displaying unique and individual characteristics. Notably, MF consistently demonstrated

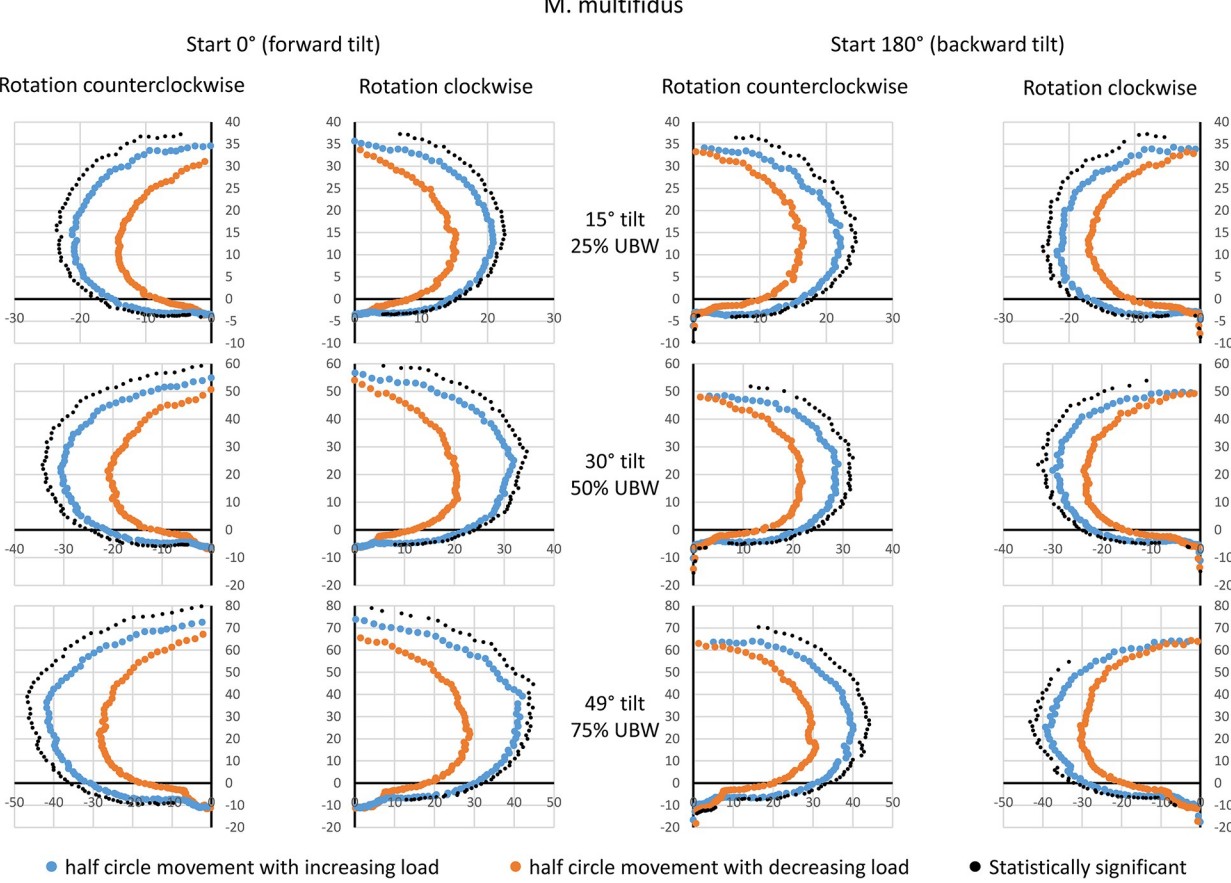

**Fig 6. Mean amplitude values during a complete 360˚ rotation for the multifidus muscle.** Data are presented based on the averaged amplitude values from both sides of the body, expressed in arbitrary units. The values for the movement half-phase with decreasing load are mirrored to enable a direct comparison between both movement half-phases. Statistical significant differences (p < 0.05, after Bonferroni-Holm correction [22]) are also displayed. UBW: upper body weight.

hysteresis, regardless of tilt angle, starting point, or direction of rotation. This suggests that for this muscle, the history of load changes is the primary factor, rather than the load level or the tension state at the start of rotation. This behavior is in line with expectations about the morpho-functional classification of trunk muscles into local and global muscles [26], or more specifically, into local and global stabilizing, as well as global mobilizing muscles [27, 28]. The MF exemplifies the functional characteristics attributed to global stabilizing muscles: control of range of motion, phasic activity dependent on the direction of movement, and notably increased activity during the eccentric movement half-phase [27, 28]. This consistent hysteresis behavior of the MF underscores its role as a global stabilizing muscle, specifically responding to changes in movement dynamics and ensuring spinal stability under varying load conditions.

In contrast, LO exhibits a characteristic similar to that observed in the abdominal muscles, with an observable hysteresis occurring only when starting from the position with load application in its main force direction [25]. For load changes starting at 180˚, no visible or detectable hysteresis was found almost for the complete rotation. However, LO displayed an atypical hysteresis at the beginning of the half-phase with increasing load, where the amplitude was lower compared to the opposing half-phase with decreasing load, unlike all other studied

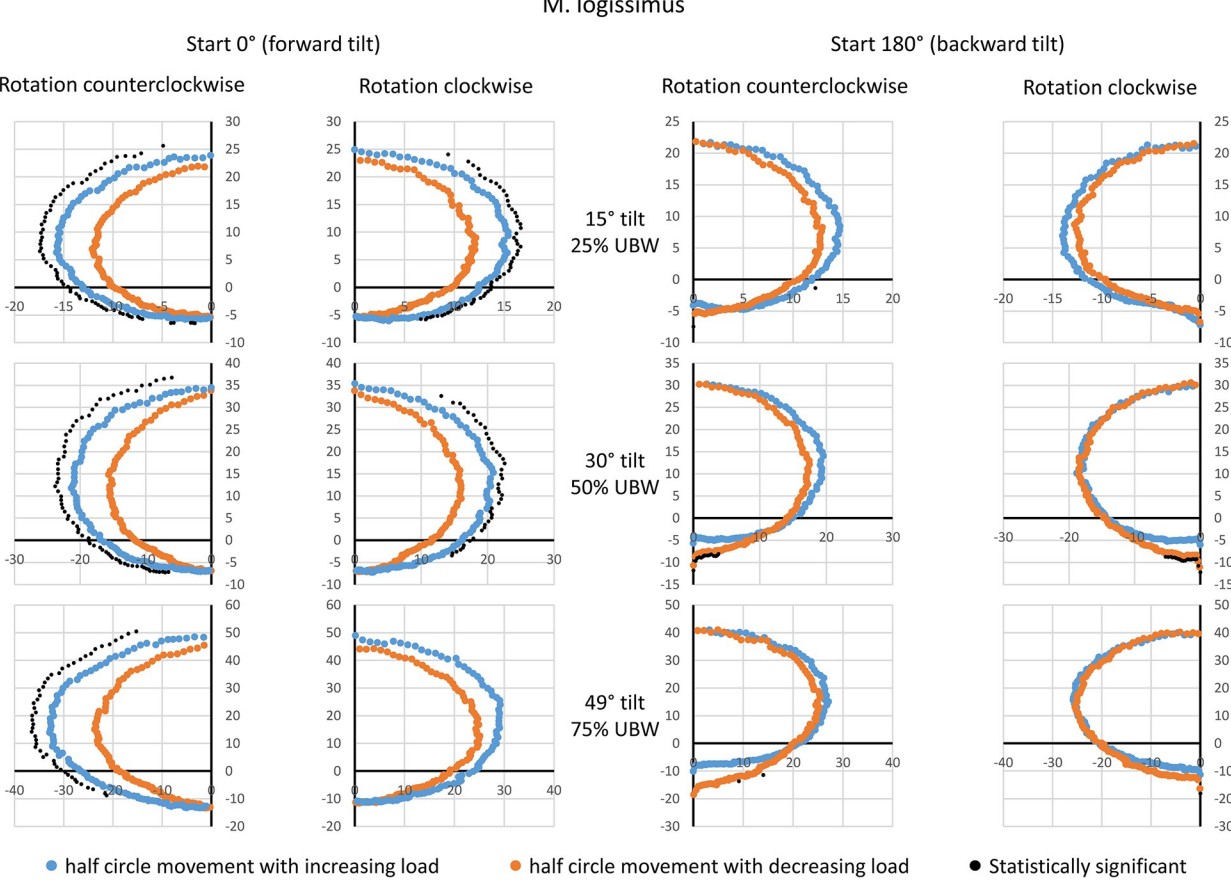

**Fig 7. Mean amplitude values during a complete 360˚ rotation for the longissimus muscle.** Data are presented based on the averaged amplitude values from both sides of the body, expressed in arbitrary units. The values for the movement half-phase with decreasing load are mirrored to enable a direct comparison between both movement half-phases. Statistical significant differences (p < 0.05, after Bonferroni-Holm correction [22]) are also displayed. UBW: upper body weight.

muscles. This suggests an increased need for stability during this phase, which likely explains the observed amplitude elevation.

The activation behavior of the ICO reveals several distinctive features. Generally, its behavior is similar to that of the abdominal muscles and LO, showing a hysteresis for the starting point at 0˚. Since the ICO by definition and location is a back muscle, one might assume that the 0˚ start point represents the main force direction. However, existing results [25] cast doubt on this assumption and is supported by the amplitude characteristics observed during the applied whole body rotation in the current study: the ICO reaches virtually equal but high amplitude values at 0˚, 180˚, and especially at ±90˚ rotational angles. To better illustrate this finding, we have superimposed selected amplitude values in comparison to constant amplitude levels (resulting in a circle, see Fig 8). From this superimposition it is evident that ICO exhibits only slight variations in amplitude across the entire 360˚ rotation, with the highest values clearly observed at ±90˚. Therefore, the starting point at 0˚ does not correspond to either the position of the highest or the lowest activation. In this sense, the defined and applied starting positions for the ICO are functionally rotated by 90˚ compared to all other muscles. Nonetheless, a hysteresis dependent on the starting position could be demonstrated. Similar to the observation of LO, an atypical hysteresis can also be observed for the ICO at the beginning of

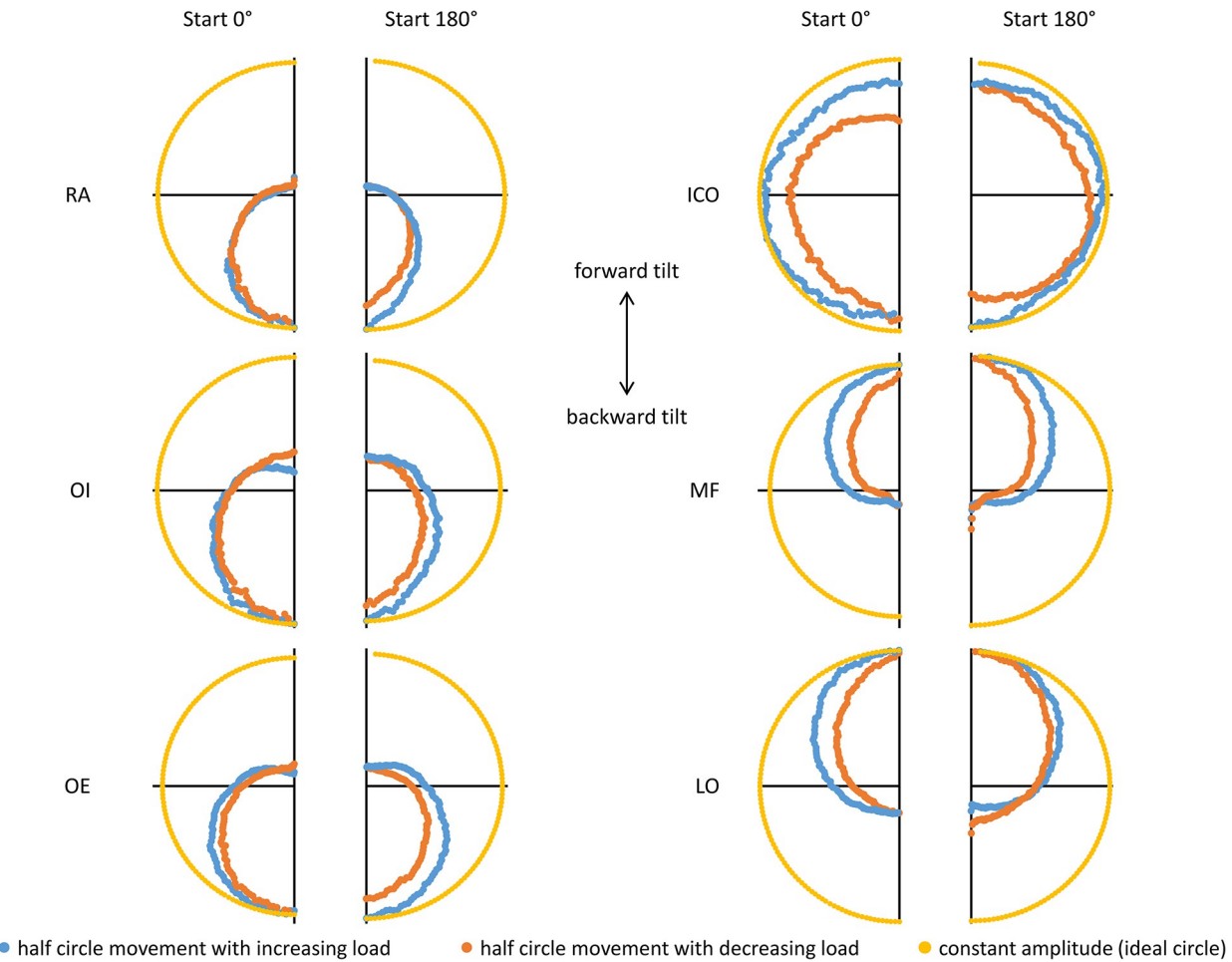

**Fig 8. Exemplary comparison of complete 360° counterclockwise rotations at 30° tilt angle (75% upper body weight) for all examined muscles, each superposed with the artificial scenario of constant amplitude.** RA: rectus abdominis muscle, OI: internal oblique muscle, OE: external oblique muscle, ICO: iliocostalis muscle, MF: multifidus muscle, and LO: longissimus muscle.

the rotation at the starting point of 180°. However, this definition remains somewhat arbitrary, as no clear increase or decrease of muscle amplitude level can be assigned to the movement half-phases.

## Limitations

The study was conducted on healthy and young individuals, which limits the generalizability of the findings. Specifically, back pain conditions could lead to differing results due to their known influence on muscular activation patterns [12, 29]. Furthermore, the data are closely tied to the use of the specific device employed in this study—whether similar results can be achieved in other examination scenarios remains uncertain. Moreover, the generalizability of the results to real-life situations involving load impacts cannot be assumed and should be further investigated.

## Conclusions

The present study demonstrated that hysteresis behavior was detectable in all examined trunk muscles. However, only the multifidus muscle consistently exhibited this behavior across all

applied test scenarios. For all other investigated trunk muscles, the occurrence of hysteresis was dependent on the initial conditions. Independent if this, no side preference could be identified. The detection of hysteresis always indicates increased activation, which is explicable through physiological mechanisms with increasing load. However, it is evident that hysteresis for almost all trunk muscles is influenced by additional external factors, particularly the activity level at the starting position. The absence of hysteresis in the relevant load change situations could potentially indicate a physiologically determined functional stability deficit, though this hypothesis needs to be validated through further research. Nevertheless, this is supported by existing evidence that unexpected asymmetrical loadings bear an increased risk of injury to the spine [30]. Specifically, our study provided experimental evidence supporting the proposed functional properties of the MF as a global stabilizing muscle, as this muscle was the only one to show the hysteresis phenomenon regardless of the external loading situation. This was not achieved for the oblique abdominal muscles, which are also defined as global stabilizers [27, 28].

## Supporting information

**S1 Fig. Effect sizes (if > 0.5) for the comparison of increasing load vs. decreasing load for the rectus abdominis muscle.** X-axes represent the respective rotation angle for the first half-phase of the complete 360˚ rotation. UBW: upper body weight.
(TIF)

**S2 Fig. Effect sizes (if > 0.5) for the comparison of increasing load vs. decreasing load for the internal oblique muscle.** X-axes represent the respective rotation angle for the first half-phase of the complete 360˚ rotation. UBW: upper body weight.
(TIF)

**S3 Fig. Effect sizes (if > 0.5) for the comparison of increasing load vs. decreasing load for the external oblique muscle.** X-axes represent the respective rotation angle for the first half-phase of the complete 360˚ rotation. UBW: upper body weight.
(TIF)

**S4 Fig. Effect sizes (if > 0.5) for the comparison of increasing load vs. decreasing for the iliocostalis muscle.** X-axes represent the respective rotation angle for the first half-phase of the complete 360˚ rotation. UBW: upper body weight.
(TIF)

**S5 Fig. Effect sizes (if > 0.5) for the comparison of increasing load vs. decreasing load for the multifidus muscle.** X-axes represent the respective rotation angle for the first half-phase of the complete 360˚ rotation. UBW: upper body weight.
(TIF)

**S6 Fig. Effect sizes (if > 0.5) for the comparison of increasing load vs. decreasing load for the longissimus muscle.** X-axes represent the respective rotation angle for the first half-phase of the complete 360˚ rotation. UBW: upper body weight.
(TIF)

**S1 Data.**
(XLSM)

**S2 Data.**
(XLSM)

**S3 Data.**
(XLSM)

**S4 Data.**
(XLSM)

**S5 Data.**
(XLSM)

**S6 Data.**
(XLSM)

**S7 Data.**
(XLSM)

**S8 Data.**
(XLSM)

**S9 Data.**
(XLSM)

**S10 Data.**
(XLSM)

**S11 Data.**
(XLSM)

**S12 Data.**
(XLSM)

## Author Contributions

**Data curation:** Christoph Anders, Leon Rosenow.

**Formal analysis:** Leon Rosenow.

**Investigation:** Leon Rosenow.

**Methodology:** Christoph Anders.

**Project administration:** Christoph Anders.

**Resources:** Christoph Anders.

**Software:** Christoph Anders.

**Supervision:** Christoph Anders.

**Validation:** Christoph Anders, Leon Rosenow.

**Visualization:** Christoph Anders.

**Writing – original draft:** Christoph Anders.

**Writing – review & editing:** Christoph Anders, Leon Rosenow.

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
