## [Decision Letter · Decision Letter 0]

19 Nov 2024

PONE-D-24-37905Studies on the hysteresis of trunk muscles - muscular specificities must be taken into accountPLOS ONE

Dear Dr. Anders,

Thank you for submitting your manuscript to PLOS ONE. After careful consideration, we feel that it has merit but does not fully meet PLOS ONE’s publication criteria as it currently stands. Therefore, we invite you to submit a revised version of the manuscript that addresses the points raised during the review process.

We look forward to receiving your revised manuscript.

Kind regards,

Holakoo Mohsenifar

Academic Editor

PLOS ONE

Journal Requirements:

3. We are unable to open your Supporting Information file:

Rotation 15 Grad 20s Start 0 Grad linksrum.xlsm

Rotation 15 Grad 20s Start 0 Grad rechtsrum.xlsm

Rotation 15 Grad 20s Start 180 Grad linksrum.xlsm

Rotation 15 Grad 20s Start 180 Grad rechtsrum.xlsm

Rotation 30 Grad 20s Start 0 Grad linksrum.xlsm

Rotation 30 Grad 20s Start 0 Grad rechtsrum.xlsm

Rotation 30 Grad 20s Start 180 Grad linksrum.xlsm

Rotation 30 Grad 20s Start 180 Grad rechtsrum.xlsm

Rotation 49 Grad 20s Start 0 Grad linksrum.xlsm

Rotation 49 Grad 20s Start 0 Grad rechtsrum.xlsm

Rotation 49 Grad 20s Start 180 Grad linksrum.xlsm

Rotation 49 Grad 20s Start 180 Grad rechtsrum.xlsm

Please kindly revise as necessary and re-upload.

Reviewers' comments:

Reviewer's Responses to Questions

**Comments to the Author**

1. Is the manuscript technically sound, and do the data support the conclusions?

Reviewer #1: Yes

Reviewer #2: Yes

2. Has the statistical analysis been performed appropriately and rigorously? 

Reviewer #1: I Don't Know

Reviewer #2: Yes

3. Have the authors made all data underlying the findings in their manuscript fully available?

Reviewer #1: Yes

Reviewer #2: Yes

4. Is the manuscript presented in an intelligible fashion and written in standard English?

Reviewer #1: Yes

Reviewer #2: Yes

5. Review Comments to the Author

Reviewer #1: Dear Editor

Thank you for the opportunity to review this manuscript. There are some specific comments:

Introduction

- Please give a reason for the particular research design in the introduction.

Methods

- The steps to derive the final sample from the initial population have not been described.

- Inclusion and exclusion criteria are not clear. Were people with hip region surgery, lumbar and/ or hip arthritis or joint inflammation, opioid analgesia, or corticosteroid intervention for pain included?

- How is the sample size determined?

- What organization or institution was the data (used in this study) for?

- Provide some references for "Investigation Procedure"

- Mention the company and country of the manufacturer of EMG

- It is mentioned in the text that "The electrode placement followed international recommendations (13, 14)."However, the cited references did not explain the location of the electrodes for the iliocostalis, multifidus, and longissimus muscles. These are deep muscles and fine-wire EMG is needed.

Discussion

- Discuss the limitations of the study, taking into account sources of potential bias or imprecision.

- Lines 338-340: "Specifically, our study provided experimental evidence supporting the proposed functional properties of the MF as a global stabilizing muscle". In this sentence, the results are generally generalized. The results of this study cannot be generalized to populations such as patients with low back pain. Rewrite the sentence according to the target population. However, in my opinion, considering that the power calculation was not done and the sample size was low, the results cannot be generalized at all.

Reviewer #2: The manuscript aimed to investigate how the activation of trunk muscles is altered under isometric conditions with continuously changing load demands.The present study demonstrated that hysteresis behavior was detectable in all examined trunk muscles. I think that this result will make important contributions to the literature.

6. PLOS authors have the option to publish the peer review history of their article (what does this mean?). If published, this will include your full peer review and any attached files.

Reviewer #1: **Yes: **Shabnam ShahAli

Reviewer #2: No

---

## [Author Response · Author response to Decision Letter 0]

25 Nov 2024

Reviewer #1: Dear Editor

Thank you for the opportunity to review this manuscript. There are some specific comments:

Introduction

- Please give a reason for the particular research design in the introduction.

Answer

We have now added a respective passage in the introduction. (L50-54)

Methods

- The steps to derive the final sample from the initial population have not been described.

Answer

All initially acquired subjects were investigated and their data went into the analysis. Maybe you mean, how the sample was acquired. As this is missing in the manuscript we have now added this particular information (L62-64)

- Inclusion and exclusion criteria are not clear. Were people with hip region surgery, lumbar and/ or hip arthritis or joint inflammation, opioid analgesia, or corticosteroid intervention for pain included?

Answer

No – we excluded any surgery in the trunk or hip region and also asked for any pain medication, which was denied by all participants. This has now been added to the respective section (L69-71)

- How is the sample size determined?

Answer

We assumed an effect size of 0.5, two sided tests, and a power of 0.8. By testing matched pairs the required sample size would have been 34 subjects. For safety reasons (taking into account technical and subject related problems) we finally investigated 40 participants. This has now been added to the respective section (L136-138)

- What organization or institution was the data (used in this study) for?

Answer

The study was no research for hire. Consequently, all collected data are the property of our institution and, ultimately, of the University Hospital Jena. At this point we are uncertain, if this particular information should be added to the manuscript. 

- Provide some references for "Investigation Procedure"

Answer

There is no specific reference for the methodology applied in this particular investigation, as these are previously unpublished data. However, extensive research has already been conducted using this device. We have now included some references. (L83-84)

- Mention the company and country of the manufacturer of EMG

Answer

We have already mentioned the manufacturer of the SEMG device. (L115). We have now also added the country for all manufacturers.

- It is mentioned in the text that "The electrode placement followed international recommendations (13, 14)."However, the cited references did not explain the location of the electrodes for the iliocostalis, multifidus, and longissimus muscles. These are deep muscles and fine-wire EMG is needed.

Answer

We have now added the missing reference.

Discussion

- Discuss the limitations of the study, taking into account sources of potential bias or imprecision.

Answer

We have now added a Limitations section (L…)

- Lines 338-340: "Specifically, our study provided experimental evidence supporting the proposed functional properties of the MF as a global stabilizing muscle". In this sentence, the results are generally generalized. The results of this study cannot be generalized to populations such as patients with low back pain. Rewrite the sentence according to the target population. However, in my opinion, considering that the power calculation was not done and the sample size was low, the results cannot be generalized at all.

Answer

As we have now added a limitations section this particular issue is mentioned there exemplarily. 

Reviewer #2: The manuscript aimed to investigate how the activation of trunk muscles is altered under isometric conditions with continuously changing load demands. The present study demonstrated that hysteresis behavior was detectable in all examined trunk muscles. I think that this result will make important contributions to the literature.

Thank you for this positive evaluation of our work.

---

## [Editor Report · Decision Letter 1]

2 Dec 2024

Studies on the hysteresis of trunk muscles - muscular specificities must be taken into account

PONE-D-24-37905R1

Dear Dr. Christoph Anders,

We’re pleased to inform you that your manuscript has been judged scientifically suitable for publication and will be formally accepted for publication once it meets all outstanding technical requirements.

Kind regards,

Holakoo Mohsenifar

Academic Editor

PLOS ONE
---

## [Editor Report · Acceptance letter]

6 Dec 2024

PONE-D-24-37905R1 

PLOS ONE

Dear Dr. Anders, 

I'm pleased to inform you that your manuscript has been deemed suitable for publication in PLOS ONE. Congratulations! Your manuscript is now being handed over to our production team.

Kind regards, 

on behalf of

Dr. Holakoo Mohsenifar 

Academic Editor

PLOS ONE